



# Imprint of Arctic sea ice cover in North-Greenland ice cores

Damiano Della Lunga[1], Maria Hörhold[1], Birthe Twarloh[1], Melanie Behrens[1], Remi Dallmayr[1], Tobias Erhardt[2], Camilla Marie Jensen[2] and Frank Wilhelms[1].

[1]Alfred Wegener Institute for Polar and Marine Research, Am Alten Hafen 26, Bremerhaven, 27568, Germany.

[2]Climate and Environmental Physics, Physics Institute & Oeschger Center for Climate Change Research, University of Bern, Sidlerstrasse 5, 3012 Bern, Switzerland.

*Correspondence to*: Damiano Della Lunga (dlunga@awi.de)

## Abstract.

Sea ice is a key component of the climate system, since it modifies the surface albedo, the radiation balance, as well as the exchange of heat, moisture and gases between the ocean and the overlying atmosphere. Hence, the reconstruction of sea ice cover before the instrumental era and the industrial times is crucial to understand the evolution of Arctic climate in the last millennium and better predict its future evolution. However, identifying relevant paleo proxies in climate archives related to sea ice cover is not straightforward. Ice cores from polar regions offer great potential to provide high-resolution records of

Arctic sea ice variability from chemical impurities such as Bromine species, which were recently proposed as indicators of sea ice extent, although their variability might be modulated by regional influences. We here use Bromine and Bromine enrichment of two ice cores form North Greenland (B17 & B26) and investigate its potential as proxy to reconstruct sea ice extent over the period 1363-1993 AD. Across the instrumental period, a good correlation is observed with the Baffin Bay and the Greenland Sea for B26 and B17 respectively, with both record showing minima corresponding to known Artic

warming events such as the 1420 AD (for B17) and 1920-1940 (Early century warming, B17 & B26), together with a strong decline starting in the late 19[th] century. We simultaneously derived a chemical classification of sea ice-related contributors of ionic species (i.e. blowing snow, frost flowers, open water) utilizing the depletion of $SO_4^{2-}$ compare to $Ca^{2+}$, $K^+$ and $Mg^{2+}$ characterizing sea ice brines and blowing snow as well the excess of $Br^-$ and $Cl^-$, characterizing frost flowers, to elucidate the evolution of the different sources. In both B17 and B26 records we observe a strong contribution of blowing snow in the

earliest part of the datasets, gradually declining in recent years in favour of open water sources.

## 1 Introduction

Sea Ice is one of the most important geophysical factors on Earth's surface and it exerts significant influence on the global climate. It can modify the radiative balance trough albedo feedbacks, reduce the ocean-atmosphere heat flux of the polar

regions and is the leading driver of 'polar amplification' which causes any climatic signal to show a larger response in polar regions (Screen & Simmonds, 2010). In the Arctic region, the 'sea-ice extent' (i.e. the integrated sum of the areas of data with at least 15% ice cover as measured by satellites), ranged from an average minimum of $6.44 \times 10^6$ $km^2$ in September to an average maximum of $15.34 \times 10^6$ $km^2$ in March between 1979 and 2018 (see 'www.meereisportal.de', Grosfeld et al., 2016; Spreen, Kaleschke, & Heygster, 2008). Over the same period, Arctic sea ice extent anomalies for each month (monthly

extent – long term mean) show decreasing trends between -2.54 % ± 0.38 (in March) and -12.76 % ± 1.91 (in September) over the course of a decade (Grosfeld et al., 2016). Furthermore, a compilation of various Arctic data sets from laser altimetry and electromagnetic sounding shows that also the average thickness of summer sea ice in the central Arctic Ocean



is declining, decreasing from >2.5 m to <1.5 m from 1980 to 2012, mostly due to the loss of thicker, older sea ice (Meier et al., 2014). Hence, it is not surprising that some studies predicted a summer with sea-ice-free conditions in the Arctic Ocean

by the late 2030s (Wang & Overland, 2012). It is therefore extremely important to unravel the natural variability of past sea-ice extent before and after anthropogenic forcing and to provide inputs to models that can correctly predict the future consequences of the strong decline we currently observe. Since direct observations on sea ice extent exist only from the 19th century and satellite measurements started only in 1979, changes in the Arctic sea ice extent must be reconstructed utilizing proxies of different nature from paleo-archives. The CLIMAP research project was the first to attempt the reconstruction of

Arctic sea-ice extent during the Last Glacial Maximum (LGM) based on the absence/presence of coccoliths, low sediment-accumulation rates and foraminifera, and the occurrence of ice-rafted debris (CLIMAP project members, 1976.). However, these results were characterized by low temporal resolution and were largely revised in the following years. Many studies since then provided further insights into past sea ice extent from coccolithophorids (Baumann et al., 2000), ostracodes (Cronin et al., 2010), planktonic and benthic foraminifera (Sarnthein et al., 2003; Wollenburg et al., 2007), ice-rafted debris

(IRD; see St. John, 2008; Darby et al., 2012), drifting wood (Funder et al., 2011) and especially dynoflagellate cysts and their assemblages (see de Vernal et al., 2013 for review). In the last decade, significant advancements towards a better understanding of past sea ice conditions were made by analysing photosynthetic biomarkers from sediment cores such as 'Ice Proxy-25' ($IP_{25}$), a monounsaturated hydrocarbon with a base structure of 25 carbon atoms, (Belt & Müller, 2013), as well as the pythoplankton marker-$IP_{25}$ ($PIP_{25}$), where $IP_{25}$ is combined with a pythoplankton marker (e.g. Brassicasterol or

Dinosterol) (Müller et al., 2011), and highly branched isoprenoids (HBI) (Fahl & Stein, 2012; see Stein et al., 2012 for a review). All the proxies mentioned above have, however, a limited (regional) spatial coverage, and a relatively low time-resolution (decadal at best). They are also limited by the activation of photosynthesis of microorganisms, which is tightly related to seasonal (first year) ice cover and it is limited in areas characterized by thick multi-year sea ice coverage. Ice core data possess the necessary requisites to overcome these limitations, and have been indeed utilized in several studies to infer

paleo-sea-ice conditions from the frequency of melt-layers (Fisher et al., 2006), and sea salts concentrations (Kinnard et al., 2006), even if the latter is highly dependent on the strength of polar atmospheric circulation and sea-salt transport (Mayewski et al., 1994). While methanesulphonic acid (MSA) in ice cores, a by-product of the oxidation of dimethylsulphide (produced almost exclusively by marine algae), has been successfully correlated with sea ice extent of some of the Antarctic sectors (Curran et al., 2003), in the Arctic this correlation is sometime inverted and strongly site-

dependent (Jonsell et al., 2007). Similarly, sea-salt-sodium records from ice cores have been positively correlated with Antarctic sea ice areas, such as the Ross Sea, over the last century (Severi et al., 2017); however, in the Arctic region, the greater dust supply and the different sources of Sodium, made the interpretation of its records more complicated (Rhodes et al., 2017). Recent ice-core studies have focused on the halogens iodine (I) and particularly bromine (Br) to reconstruct the extent of sea ice in several specific sectors of the Arctic region from decadal (Spolaor et al., 2016a) to centennial (Maselli et

al., 2017), to glacial-interglacial time-scales (Spolaor et al., 2016b) . Here we present an attempt to provide further ice-core based reconstruction of the relative sea-ice extent in the Arctic sector surrounding Greenland for the period 1502-1993 AD based on the profiles of Bromine and other ionic species from two different ice/firn cores of the North Greenland Traverse ice core array (B26 and B17, see fig. 1). The chemistry of bromine in relationship to the sea ice environment is discussed in paragraph 1.2. Furthermore, we provide here the simultaneous investigation on the contribution of the different sea-ice

constituents (e.g. blowing snow, frost flower), as well as from open water, to the ionic budget of Greenland ice cores. In order to elucidate this sea-ice imprint on ice-core ionic species, we utilize de-trended profiles of. $Na^+$, $SO_4^{2-}$, $Mg^+$, $Ca^{2+}$, $K^+$, $Cl^-$, which are essentially related to the brines residing on the top of forming sea ice and to the environments in the immediate surroundings, as discussed in the following section (§ 1.1).



### 1.1 Sea ice imprint on ionic chemistry

Ice and sea salts do not form a solid solution and therefore, during the formation of sea ice, ions do not enter the solid phase. In a classic representation of phase relations in sea ice the 'standard' suit of components includes ions such as $Na^+$, $Ca^{2+}$, $Mg^{2+}$, $K^+$, $Cl^-$, $SO_4^{2-}$ and $CO_3^{2-}$ (Assur, 1960), representing around 99% of the salts in sea ice by weight. These are followed by bromide, fluoride, boron and strontium that make up most of the remaining 1% in weight. During a process called 'brine rejection' all of these ions are rejected from the lattice of the forming ice and accumulate in the surrounding seawater,

creating saltier, denser brine. Some droplets are entrapped in ice crystals creating pockets of brine that remain in liquid state as much cooler temperature would be required for it to freeze. The brines can migrate downwards by gravity or drainage during melting events, but also move upward in the forming sea ice by capillarity action, creating a salty liquid layer residing on top of the newly formed ice (Griewank & Notz, 2013). When air temperature cools down below freezing point (-1.86 °C for 34 practical salinity units [PSU] ocean water), ice is forming. Proceeding with further cooling, we observe precipitation

of new mineral phases within the brine pockets, since the brines become supersaturated relative to those salts. At -8.2 °C, the brine is supersaturated relative to sodium sulphate, triggering the precipitation of mirabilite ($Na_2SO_4 \cdot 10H_2O$). This reduces the sulphate and sodium content in the residuals liquid phase, but also their relative ratio ($SO_4^{2-}/Na^+$), which decreases as temperature in the residual brine gets lower and further precipitation is occurring. Other salts precipitating during the freezing of seawater may include ikaite ($CaCO_3 \cdot 6H_2O$), which can precipitate, in the absence of the formation of anhydrous

$CaCO_3$ and if equilibrated with current $CO_2$ level, already at -4.5 °C. However, Ikaite formation is directly dependent on phosphate concentration, which are lower in the near-Greenland seas than in Antarctic seawater and it has been rarely observed in the Arctic sea ice (Dieckmann et al., 2010; 2008). At -22.9 °C, approximately 90% of precipitable sodium chloride is removed from the brine in the solid form of hydro-halite ($NaCl \cdot 2H_2O$). This also introduces a depletion of $Na^+$ & $Cl^-$ compared to $Mg^+ + K^+ + Ca^{2+}$, whose salts precipitate only at much lower temperatures and are seldom encountered over

Arctic sea ice.

Blowing snow and frost flowers can wick up the brines from the fresh sea-ice surface via capillary action. When snow gets mobilized by wind the brine that was attached to the snowflakes via absorption is transported together with the precipitated phases adhered to the snowflakes.

Frost flowers only form in rather calm environment, where the brine that resides on top of the sea ice surface has time to

sublimate, creating a 1–3 cm thick water vapour layer just above the surface that is supersaturated with respect to ice. In this layer, any crystal growth is therefore enhanced, resulting in the growth of frost crystals. Underneath the crystals, a saturated brine layer forms and increases in thickness up to of 2–4 mm typically. In the end, the surface tension effects draw up the surface brine onto the frost crystals, producing the large salinities observed in frost flowers (Thomas, 2017).

The salinity of frost flowers is also almost twice than the one observed in the brines and they typically show a higher Cl/Na

ratio and a stronger bromine excess (Roscoe et al., 2011).

Given the fact that temperature over peripheric Arctic sea ice (as the one surrounding Greenland) ranges, on average, between -28 °C in March to -1 °C in July (with values between -15 °C and – 4 °C from April to June) (Lindsay & Rothrock, 1994), all the fractionation mechanisms described in this paragraph are likely to characterize aerosol originating from the sea ice environment and have been indeed already observed in polar regions (Domine et al., 2004; Rankin et al., 2002, 2004;

Wolff et al., 2003). Based on this, we introduce in section 2.5 a classification diagram to elucidate the evolution over time of the contribution of sea ice-blowing snow, frost flowers and open water to the residual ionic budget of B17 and B26.

### 1.2 Sea-ice bromine

The main source of inorganic bromine in the marine boundary layer (MBL) is the ocean water, where $Br^-$ is present in concentrations of less than 67 ppm. Following 'brine rejection' during the freezing process, however, pockets with high

concentration of ionic species, including $Br^-$, are found on the surface of fresh sea ice (Saiz-Lopez et al., 2012).



When sunlight hits the surface of sea ice, molecular bromine produces atomic bromine radicals through photolysis, and subsequently forms bromine monoxide, BrO, through the reaction with ozone:

$$Br_2 + hv \rightarrow 2Br\cdot \qquad (\lambda<380 \text{ nm})$$

$$Br\cdot + O_3 \rightarrow BrO + O_2$$

Self-reaction of BrO may form two bromine atoms (85 %) or a Br molecule (15 %) which is again photolyzed, leading to a mechanism of catalytic behaviour that destroys ozone and exponentially increase concentration of Bromine in the gas phase (Wennberg, 1999):

$$BrO + BrO \rightarrow Br\cdot + Br\cdot + O_2$$

$$BrO + BrO \rightarrow Br_2 + O_2$$


The net effect of such reactions, known as *bromine explosion*, is to produce oxygen ($O_2$) and bromine species (HBr, Br) at the expenses of ozone ($O_3$). In springtime, when the amount of solar radiation in the arctic region begins to increase, photolysis triggers the *bromine explosion* leading to the deposition of HBr and Br on the snow surface, once $O_3$ is depleted sufficiently. Satellite measurements confirmed that a pronounced springtime/summertime increase in gaseous BrO

concentrations is observed over sea ice in Antarctica (Schönhardt et al., 2012) and the Arctic (Begoin et al., 2010) in recent times. This is related to the fact that the brines residing on top of first-year sea ice offer the critical concentration of Bromine that is required to initiate the autocatalytic reaction of the *bromine explosion*. A recent 1-D chemistry model simulation predicted an increase of bromine deposition on surface snowpack over Sea Ice after 24 to 48 h of recycling over first-year sea ice (Spolaor et al., 2016a). The stability of the bromine in the snowpack was investigated both at Summit station,

Greenland (Thomas et al., 2011) and in East Antactica (Legrand et al., 2016) revealing in both cases, that the snowpack cannot account for the observed gas-phase inorganic bromine in the atmosphere. Bromine enrichment in snow (compared to sodium, relative to seawater) has been recently used to better quantify the sea ice variability from ice cores both in the Antarctic and Arctic regions (Spolaor et al., 2013a, 2016a) and it is defined as:

$$Br_{enr} = \frac{Br^-}{(ssNa^+ \cdot 0.00623)}$$


Where sea salt sodium ($ssNa^+$) is defined as in (Röthlisberger et al., 2002):

$$ssNa^+ = \frac{(1.78\ Na^+) - Ca^{2+}}{1.818}$$

To quantify how much Bromine is produced during the *bromine explosion* it is also useful to define a bromine excess value:


$$Br_{exc} = Br^- - \frac{ssNa^+}{0.0623}$$

Bromine, and especially bromine enrichment, are therefore directly proportional to the extent of the surface available for the *bromine explosion* and thus can be utilized as an indicator of local sea ice extent variability in the Arctic region. Although the *bromine explosion* is predominantly occurring on first-year sea ice, it can sometimes occur on multi-year sea ice, since

the average salinity of sea ice is generally higher than seawater up to a thickness of about 3 m, and this can be sufficient to trigger to catalytic reaction cycle (Cox & Weeks, 1974).



## 2 Data analysis and methods

### 2.1 North Greenland Traverse ice cores

We present here data from two different ice cores (B17, B26) drilled during the North Greenland Traverse campaign
between 1993 and 1995 (fig. 1). Table 1 provides an overview of the characteristics of the different drilling sites. Data of
major cations ($Ca^{2+}$, $Mg^{2+}$, $K^+$ and $Na^+$) and anions ($Cl^-$, $SO_4^{2-}$, $Br^-$, $NO_3^-$) from core B26 and B17 were obtained at
University of Heidelberg and Alfred Wegener Institute (AWI) by ion chromatography in 1996 and 2018 respectively
(Sommer, 1996; Schwager 2000). For B26 only data down to a depth of 88.4 m [i.e., AD 1501] are available in a yearly
resolution. Details on the methodology are given in Sommer (1996) and Schwager (2000) and are summarized in the
supplementary information. The 100.8 m B17 ice core was analysed at the University of Bern in 2017 by Continuous Flow
Analysis (for details on the method see: Kaufmann et al., 2018; Erhardt et al., 2019; Burgay et al., 2019 and references
therein), coupled with a custumized fraction collector (BESTA-Technik GmbH, Germany) sampling the melt-water every
40-60 s. Details on the methodology are reported in the supplementary information.

### 2.2 Calculation of fluxes

The impurity content of the NGT ice core array is related to accumulation rate by a weak direct linear relationship,
characteristic of each element ($r^2$=0.13-0.32, see fig. s1), whereas the variability of each species is inversely proportional to
accumulation rate (fig. s2 and Weißbach et al., 2016). We here define the variability as the average amplitude of the
residuals from baseline correction achieved via asymmetric least square smoothing (ALSQ) as in (Peng et al., 2010) and
performed for each individual time series . In order to account for these effects, we display profiles as fluxes ($F_i$) of species
(i) in the corresponding ice core (x):

$$F_{ix} = C_{ix} \cdot Acc.rate_X$$

Where $C_i$ is the concentration (in ppb) of the species (i) and where the profile of accumulation rate for each core have been
obtained multiplying the layer thickness in m (after layer-thinning correction based on Nye model; see supplementary
information, fig. s3) by the density in kg/m$^3$ as given in Wilhelms, (1996). The layer thickness was inferred by annual layer
counting performed on the depth-profiles of several chemical species ($Ca^{2+}$, $Mg^{2+}$, $K^+$ and $Na^+$), conductivity and $SO_4^{2-}$.
The chronology for both cores was refined by multiple-proxies layer counting performed on the basis of the year-counting
presented in Weißbach et al. (2016). Figure 2 shows the original bromine record of B17 and B26 while figure 3 presents the
bromine enrichment for both cores whereby the B17 record has been down-sampled from the original resolution to annual
resolution to match the B26 ice core record.

### 2.3 Back-trajectory modelling

In order to infer the major source areas of chemical species deposited in Central Greenland in recent times, back-trajectories
have been calculated with the HYSPLIT model (HYbrid Single-Particle Lagrangian Integrated Trajectory, v. March 2019)
using NCEP/NCAR, Reanalysis data from the National Weather Service's National Centre for Environmental Prediction
provided by the NOAA's Air Resources Laboratory (Draxler & Hess, 2009), (fig. 4). The arrival heights have been chosen
according to values in table 1. Trajectories were calculated on total air masses moving across each 1° x 1°-degree lat-long
cell each 12 h using vertical motion mode for a total run duration of 12 days and merged over the 3-month spring period of
March-April-May (MAM) and summer period of June-July August (JJA) between 1950 and 1990 AD and expressed as
frequency (number of trajectories passing through each grid square / number of total trajectories [each trajectory counted
once per grid cell]). A 12 days run time was chosen according to estimation of hydrogen bromide maximum residence time
in the atmosphere (Wofsy, et al., 1975).





### 2.4 Comparison of Bromine with Sea ice concentration

Bromin enrichment of B17 and B26 have been averaged together ('stack') over the time period 1501-1993 AD and normalized between 0 and 100, to be subsequently correlated with the Arctic sea ice extent reconstruction from Kinnard et al.( 2011) in fig.5. In order to further validate the record and to infer the source areas and the zones of most influence of B17

bromine enrichment, maps were generated (fig. 6) point-correlating bromine enrichment for both proxies with HadISST sea ice concentration on a 1-degree latitude-longitude grid between 60°N and 90°N for the time interval 1800-1993 AD (Rayner et al., 2003). These datasets have been down-sampled to a resolution of one single datapoint per year obtained averaging monthly values within the March-August period following the indications from proxy seasonality (fig. s4) and back-trajectories (fig. 4), in order to minimize the time-resolution differences to our ice core data.

### 2.5 Chemical signature of Sea Ice components and open water

In order to identify the relative contribution of blowing snow, open water and frost flowers as the primary source of bromine and other sea-ice related ions, we constructed ternary diagrams for B17 and B26 (fig. 7) with indexes defined as follows, based on the chemistry described in section § 1.1:

i.   A measure of depletion of sulphate compared to total magnesium, calcium and potassium (conversely, it can be expressed as a relative enrichment of the mentioned cations compared to $SO_4^{2-}$), indicative of a 'brine'- enriched signal carried by blowing snow or ('$B.Sn$'):

$$B.Sn = \frac{res(Ca^{2+}) + res(K^+) + res(Mg^{2+})}{res(SO_4^{2-})}$$

ii.  A measure of frost flower contribution, given by the multiplication of Br excess and Cl excess:

$$FF = res(Br_{exc}) \cdot res(Cl_{exc})$$

$$where \ Cl_{exc} = [Cl - 1.80Na];$$

iii. A measure of sea salt contribution from open water 'O.W':

$$OW = \frac{1.164Na}{Cl}$$

Where *res ()* indicate residuals after baseline subtraction performed by ALSQ smoothing (asymmetric factor= 0.001; threshold=0.05; smoothing factor=4; n° iterations=10), utilized here to avoid bias from long-term trends. The diagram is

based on the assumptions that: (i) blowing snow residing on top of sea ice will carry an enriched $(Mg^+ + Ca^{2+} + K^+)/ SO_4^{2-}$ signal from the absorbed brines, (ii) A frost flower related signal will carry a high Br and Cl excess, being enriched in these values; (iii) A seawater related signal will carry a $Cl^-/Na^+$ value close to 1.164 (seawater). All time-series of residuals have been normalized between 0 to 100 to provide equal weights and the diagram have been constructed as contour maps where coordinates of data point have been scaled so that the sum *B.Sn* (i) + *FF*(i) + *OW*(i) would be 100.

## 3 Results

### 3.1 Fluxes of Bromine species

Bromine fluxes for B17 and B26 are shown in fig.2. Across the entire profile, B26 presents an average value of 87.4 pg/m²a compared to 48.3 pg/m²a for B17, therefore fluxes for the latter are roughly 55% of the values of the former. The absolute





minimum for both ice cores resides in the 20[th] century, specifically 1940 AD and 1960 AD for B17 and B26 respectively,
with values between 9-14 pg/m$^2$a. Both profiles present then a rise in the second part of the 20[th] century where fluxes rise
back to 50 pg/m$^2$a, before significantly decrease again from 1985 AD. The B17 ice core present a first minimum around the
years 1420 AD, which may be related to a period of relative warming in the Arctic (Weißbach et al., 2016). This is followed
by a period of relative high fluxes in the B17 ice core, between 1450 and 1520 AD, and a century of relatively lower values
between 1520 and 1620 AD for both B17 and B26. The decreasing trend has been investigated in both cores utilizing a
RAMPFIT (Mudelsee, 2000) regression with initial guess of ± 30 years on the transition points. We can observe that the
initiation of the decreasing trend starts early already at the beginning of the 18[th] century for both cores but the decrease is
significantly more marked in the B26 ice core. The range of uncertainty has been quantified by normal equi-tailed (1-2α)
bootstrap confidence intervals with 2000 resamples and α=0.025 and are highlighted by the yellow boxes in fig.2 and 3.
Fluxes of bromine enrichment are shown in fig. 3. In this case, the initiation of the decreasing trend appears much later in
time, respectively around 1870 and 1860 AD for B17 and B26 respectively. This difference might be related to a relative
decrease of $ss$Na$^+$ concentration in the period 1700-1900 AD (see fig. s5), which maintained the constant level of bromine
enrichment observable till the late 19[th] century. Both profiles show two clear intervals of low values between 1600 and 1630
AD and between 1650 and 1670 AD where values decrease of roughly 60-80% for B17 and B26. The values are then stable
in both B17 and B26 till the second part of the 19[th] century but are punctuated by simultaneous minima (e.g. 1737 AD) and
maxima (e.g. 1755 AD).
The initiation of the decreasing trend lies for both cores within the confidence interval of the RAMPFIT model, immediately
before a maximum situated for both cores around the year 1875 AD. The minimum of the entire profile is situated for both
B17 and B26 between 1920 and 1940 AD, corresponding to the well-known period of Arctic warming known as 'Early
Twentieth-Century Warming' (Yamanouchi, 2011, Weißbach et al., 2016).
This is followed by a period of generally increasing fluxes until the 1980s when the values start to decrease again.

**3.2 Origin of Air masses**

Back-trajectories are shown in fig. 4 and indicate possible areas of origin of the air masses responsible of the aerosol
transport to the B17 and B26 sites. The seasonality of proxies and especially bromine signals has been investigated in fig. s4
and clearly show a maximum of bromine activity in Spring and Summertime, in agreement with previous findings (Spolaor et al.,
et al., 2016a, Maselli et al., 2017). Thus, we will focus hereafter on back-trajectories for this part of the year. In Springtime
(MAM) air masses arriving at the B17 site originate from both East and West side of Greenland, and rarely from the north.
The most significant contributor seems to be in this case the Greenlandic sea, which is covered by a sea ice in the most
proximal regions between 0.1 and 0.4x10$^6$ km$^2$ (Germe et al., 2011). Many air masses, however, seem to originate from the
Baffin Bay just west of Greenland and rapidly move eastward to reach the B17 site. Finally, a few trajectories seem to
originate north, in the Arctic ocean and very rapidly move south, but this contribution seem to be negligible.
Similarly, B26 springtime-trajectories include possible sources in the Arctic ocean, but these are originating at more
proximal location such as the Nares or Lincoln Seas and seem not preponderant. The most important originating area is again
the Baffin Bay and all the coastal location on the West side of Greenland, as far south as the Davis Strait and the Labrador
Sea. The Sea between Iceland and the West coast of Greenland (Denmark Strait) seem to contribute partially to air masses
arriving at the B26 site, but these seas have a sea ice coverage that is very limited to coastal sites (Germe et al., 2011),
whereas the Baffin Bay area have a strong Sea Ice coverage and variability between 0.2 and 0.8 10$^6$ km$^2$ (Cavalieri et al.,
1996).
In the summer period (JJA) this picture is slightly modified, and we observe almost exclusively air mass coming from East,
as the Greenlandic and even the Norwegian Sea in the case of B17, and an increase in the frequencies over the Baffin bay
and the Canadian Arctic for air masses reaching the B26 location. Following that: (i) the greatest Sea Ice variability occurs in



the Baffin Bay area for both Spring and Summertime and, less strongly, in the Greenland sea; (ii) Bromine explosions occur over first year sea ice and therefore are proportional not only to the Sea Ice extent but also to its variability between winter and Spring/Summer; (iii) back-trajectories shows that originating areas are mostly the Greenlandic Sea and the Baffin bay areas, we consider that B26 record will have a strong influence from the Baffin Bay Sea Ice extent, while the B17 will be mostly reflecting variation of Sea Ice in the Greenlandic sea.

### 3.4 Variability of Sea Ice extent

We compared the variability of Bromine-enrichment fluxes of B17 and B26 with different sea-ice extent reconstructions. Figure 8 shows the Sea Ice extent (in millions of $km^2$) for the Baffin Bay area and Greenland Sea as inferred from NSIDC monthly sea ice concentration (Cavalieri et al., 1996) for the period 1980-1995. The Baffin Bay Sea Ice extent clearly show a strong intra-annual variability and an overall greater area covered by sea ice if compared with the Greenland Sea, both in winter and in summer. The interannual variability is weaker and we can observe only a significant decrease starting in the AD 1990. This is also mimicked in the B26 Br_enr which show a concomitant two-fold decrease over a single year, to reach a minimum in AD 1993. Based on Back-trajectories most frequent pathways, we choose to compared B26 with the Baffin Bay sea ice extent and the two profiles show a moderate correlation (r=0.208, p<0.5), when performed between the B26 Br_enr and the 12_points Savitsky Golay moving average (to account for resolution discrepancies). In the case of B17, the correlation between the 12-points Savitsky Golay moving average and the B17 Br_enr is slightly higher (r=0.271, p<0.35). Both curves show a decreasing trend in the fluxes between 1980 and 1985 followed by a gentle increasing trend between 1980 and 1990 AD, before dropping again in 1990 AD. This seems to indicate that Br_enr retain some capabilities to record the variability of Sea Ice extent of certain regions on a year to year scale.

Figure 7 show maps of B17 and B26 Bromine enrichment fluxes correlated with Sea Ice cover from the HadISST database for the time interval 1800-1993 AD (Rayner et al., 2003). The correlation is expressed as Pearson correlation coefficient between -1 (perfectly anti-correlated) and 1 (perfectly correlated). In the case of B17, the correlation is quite strong (0.6 to 0.7) with the entire eastern sector, especially with the central Greenlandic Sea and the Seas around Svalbard as well as from coastal ice near the Southern part of Greenland. Interestingly, no significant correlation emerges between the B17 Bromine enrichment flux and the Baffin Bay area, neither the Canadian Artic nor the Arctic Sea north of Greenland, which appear also anti-correlated in some sector, as the Davis strait (-0.2 to -0.3). In the case of B26, the map shows a maximum located in the Labrador sea and the Davis strait and generally the Baffin Bay area (0.4 to 0.5), whereas the Eastern sector appear only locally correlated (only coastal areas) and overall slightly anti-correlated in the Greenland sea (-0.1 to -0.3).

Since the correlation map presents differences possibly due to regional influences of sea ice from Baffin Bay and Greenland Sea for B26 and B17 respectively, we investigated the Pan-Arctic correlations of our records, by comparing the historic Arctic sea ice reconstruction of Kinnard et al. (2011), which is based on a network of high temporal resolution proxies with large spatial coverage, and the normalized average of B17 and B26 Bromine enrichment (fig. 5), on the assumption that the latter might be representative of the variability of a larger Arctic sea ice area compared to the individual records. These two profiles are shown in fig. 5 which can be divided in three different segments. Between the AD 1500 and 1800 the correlation is strong (r=0.398, p<0.01) but decreases significantly in the following years until approximately AD 1890 (r= 0.0919, p<0.02). In the most recent section, between AD1890 and 1993, the correlation is again significant (r=0.280, p<0.02).

The profiles in fig. 5 show simultaneous minima immediately before and after the year 1600 AD, which are consistent with low sea ice coverage reported for the Canadian Arctic (Richerol et al., 2008) and the Chukchi sea (McKay et al., 2008) in this period and generally a high sea surface temperature anomaly (Kinnard et al., 2011). Starting from the year 1800 AD our average of bromine enrichment and the reconstructed Arctic sea ice coverage from Kinnard et al. (2011) start to diverge, until the early 20th century were trends are again very well correlated and show an unprecedented drop in recent years, mimicking the sea ice decline of the modern era.





This discrepancy might be explained by a combination of different factors. A general increase of the dust fluxes of size> 1.2 µm at the beginning of the 19th century (fig. s6) is observed in the B17 record. In this range of airborne particles, it has been

frequently observed a large bromine depletion compared to sodium, due to fast recycling of bromine species facilitated by larger reaction areas (Yang et al., 2008). A relative increase in dust of larger sizes could therefore result in a depletion of Bromine compared to sodium and trigger the decrease in Br_enr observed, despite the concomitant increase in acidity (particularly $HNO_3^-$, see fig. s6) which would increase the solubility of Bromine species. This, however, is limited by the fact that the depletion of Br compared to Na is mostly observed in sea salt particles which are marginally detected during

continuous flow analysis as they are likely dissolved into the meltwater.

More importantly, It has been also observed, that the sea ice coverage nearby the coast of East Greenland dramatically decreased at the beginning of the 1830s based on observation on tree rings and ice cores-$\delta^{18}$O from the Arctic (Macias Fauria et al., 2010). It is therefore possible that the different trends reflect solely a regional effect characterizing both B17 and B26 records which were dominated in the early 19th century by air masses originating in the Greenland Sea (Maselli et al., 2017).

**3.5 Contribution of Blowing Snow, Frost Flowers and Open Water**

Figure 7 shows the ternary diagram for B26 (top) and B17 (bottom) relating the contribution of the chemical signatures of Blowing Snow (B.Sn), Frost Flowers (F.F) and Open Water (O.W), as defined in section 2.5. The diagrams are drawn as contour maps with colours indicating time from 1500 AD (blue) to 1993 AD (brown).

In the case of B17, we notice that most of the early years up to 1650 AD (blue colours) reside in the top corner, which

indicate a strongly enriched $(Mg^+ + Ca^{2+} + K^+)/ SO_4^{2-}$ ratio. Between 1650 and 1750 AD (blue-green colours) most of the data points fall into the central part of the diagram, indicating a decrease in this depletion of Sulphate and a relative increase in Br_exc and Cl_exc and a possible stronger input of ocean sea salts. In the period from 1850 to 1993 AD all the values reside on the bottom line and, more significantly, in the bottom right corner, indicating a strong predominance in the signal of $Na^+$ and $Cl^-$ from sea salts.

This seems to indicate the predominance of blowing snow as ionic source in the early part of the dataset, declining over time in importance in favour of the open ocean source. Frost-flower contribution seem to have a positive trend with time, but high and low values of F.F are distributed more heterogeneously over time and we therefore consider those subordinated to the major trend from blowing snow to open ocean.

In the case of B26 a similar trend is observed, but the pattern is somehow less regular, and we can observe some high values

for blowing snow as well as for frost flower in the most recent years (1850-1993 AD). We can observe that the earliest datapoints (blue colours) reside still on the top of the diagram whereas most of the recent years are indeed confined to a 'limited blowing-snow scenario', outlining the increasing trend of $(Mg^+ + Ca^{2+} + K^+)/ SO_4^{2-}$. Again, Frost Flowers show a pattern of clear increase over time.

Overall, the diagrams seem to indicate a change of the chemical signature over time from Blowing Snow to Open Water (and

possibly increased formation of Frost Flowers), which is compatible with a relative increase of Open Water surface at the expenses of Sea Ice with time. It cannot be excluded, however, that the increase of $(Mg^+ + Ca^{2+} + K^+)/ SO_4^2$ ratio, which characterizes both contour maps, reflects a decrease in the variability of the $SO_4^{2-}$ introduced by anthropogenic sources, even after detrending of B17 and B26 time series (which should remove longer trends in dust variability). On the same note, high values of Frost Flower source (F.F) in recent years can be explained by an increase in the Br_excess and Cl_excess in the

most recent part of the time series, immediately after the minimum which characterizes the arctic warming interval around 1940 AD (fig. s4).



## 4 Discussions and conclusions

Gaseous bromine species (such as BrO) in the Arctic boundary layer have concentrations that range from few ppt to several tens of ppb, with approximately a two-fold increase in concentrations during summer known as '*bromine explosion*' which
have been observed extensively in satellite measurements in recent years in the Arctic region (Richter, et al., 1998; Begoin et al., 2010; Hollwedel et al., 2004). The bromine explosion consists in a series of autocatalytic chemical reactions which lead to an increase in the production of bromine species in the atmosphere residing on top of sea ice. This reaction chain produces a peak in concentrations around 4 to 6 days after the initiation of the bromine explosion by photolysis, but an increase in concentration in the deposited snow can be observed already 24 to 48 hours after recycling on the top of sea ice (Spolaor et
al., 2016a). However, the bromine explosion process occurs only when an already enriched bromine source is present since a critical concentration of bromine is required to initiate the autocatalytic process by photolysis (Simpson et al., 2007).

For this reason, the summertime bromine explosion is mostly related to first-year sea ice, where brines are still largely present on the ice surface and were not drained through the ice pack as in multiyear sea ice (Simpson et al., 2007).

This seasonality in the bromine signal has been observed in several ice core record from the Arctic (Spolaor et. Al, 2016a;
Maselli et al., 2017) and it is confirmed in our B17-B26 records (fig. s4). The relative phasing of our proxies suggest that $ssNa^+$ peaks in winter as a result of increased storminess over the Arctic region, followed by a springtime dust peak ($nssCa^{2+}$) and a late-spring acidity ($nssSO_4^{2-}$, $NO_3^-$) maximum, immediately before the summer peaks in Br, Br_enr and Br_exc. This is compatible with reported seasonality from ice core record (Maselli et al., 2017), with Satellite measurements (Begoin et al., 2010) and modelling studies (Yang et al., 2005), which show a Bromine and Bromine enrichment maximum
during the summer months despite the decrease in atmospheric BrO flux observed over summer induced by the reduction of young sea ice and ice salinity due to melting (Maselli et al., 2017). The discrepancy is probably a result of the lag between bromine-containing particles becoming airborne and thus allowing bromine to become enriched above the seawater ratio compared to $Na^+$, and the final deposition (Spolaor et al., 2014).

Overall, both Bromine and Bromine enrichment record for B17 and B26 show profiles with similar trends (fig. 2 and 3): In
the case of bromine fluxes we observe synchronous minima immediately after 1600 AD and during the interval 1920-1940 AD, an increasing trend in the last 50 years of the record (1940-1985 AD), and a rapid drop in the years between 1985 and 1993 AD. The initiation of the trend can be pinpointed as far back as the beginning of the 18[th] century.

Similar features are observed for Br_enr profiles; however, the initiation of the decreasing trend appear later in time, at the end of the 19[th] century. This is due to a relative decrease in the $Na^{2+}$ concentration in the interval 1750-1900 AD concomitant
to the decrease in bromine, leading to constant values of Br_enr until roughly 1875 AD.

The averaged curve of Br_enr, as a method to capture the underlying mode of variability that link both records and compensate for regional effects, shows that the B17 and B26 records are very well correlated with Arctic sea ice extent reconstruction already available (Kinnard et al., 2011), at least for the pre-industrial part.

After 1875 AD, the 'stack' of Br_enr and the reconstruction from Kinnard et al. (2011) show diverging trends. The reason of
this feature might lie in the fact that an increased levels of dust input, as observed in the B17 ice core, introduces a high amount of particles of relative bigger size (diameter> ~1 μm), which offer a reaction substrate that promotes the exchange and the net loss of bromine species to the atmosphere (von Glasow et al., 2004). Laboratory experiments have shown that large bromine depletion (between 10 to 100%) occur in aerosol particles of size bigger than 1μm (Enami, et al., 2007), so this effect might overcome also the increase in Bromine mobility that should result from the increase in acidity typical of
industrial times (see fig. s4, especially $NO_3^{2-}$). On the other hand, it is possible also that the increase in acidity would promote Bromine loss from particles and snowflake both in the aerosol and in the post-depositional phase, allowing bromine species to be recycled further and be transported further away from the depositional sites of the Greenland ice sheet and back to the ocean. However, this would probably have a minor or local effect, as most of the loss of Bromine compounds would be compensated by eventual wet and dry deposition to the snow surface (von Glasow et al., 2004).



HBr can be also emitted by volcanoes, producing spikes in concentrations of several tens of ppb but accounting globally for less than 0.1% of Bromine compared to sea salt on the background level (Pyle & Mather, 2009). These are therefore not investigated further here, even if is interesting to point out a spike in both the B26 and B17 Bromine enrichment flux around the year 1883 AD, when the Krakatoa eruption occurred.

We can observe that the initiation of the decreasing trend of Br_enr roughly correspond with the termination of the Little Ice Age, which have been observed between 1420 and 1850 AD for North Greenland ice cores (Weißbach et al., 2016). Maselli et al. (2017) also observed a sea ice decline, inferred by MSA and Br proxies, in the Summit and Tunu ice cores, starting around the middle of the 19$^{th}$ century and preceded by a low sea ice event around 1830 AD, which has been also observed in other Arctic sea ice multiproxy reconstruction as well (Macias Fauria et al., 2010). This has been related to a decline in sea ice extent along the eastern coast of Greenland and the Baffin Bay (Macias Fauria et al., 2010), and suggests that the B26 and B17 Bromine might have travelled from proximal sites west and east of Greenland, respectively, and therefore declining together with its sea ice extent. Our trajectory maps and sea ice correlation maps (fig. 4 and 6) confirm that the Greenland Sea is the area of major influence for the B17 site under current climate conditions. This is also corroborated by the good agreement of the NSIDC monthly sea ice concentration and the B17 Br_enr (r=0.271, p<0.05), even on a relative short time window (1980-1993 AD), and the correlation map of B17 Br_enr and the HadISST sea ice concentration database for 1800-1993 AD, which clearly illustrate a strong link between our record and all the Greenland Sea.

In the case of B26, the good correlation of our Br_enr with the Baffin Bay sea ice variability in the same recent time (r=0.208, p<0.05), together with back-trajectories clearly showing a western area of origin, suggests that bromine enrichment values are effected mostly by sea ice variability of the Baffin Bay.

It is interesting to notice that the area immediately around Greenland with higher concentration of BrO in the atmospheric column reported by GOME-2 BrO (Spolaor et al., 2014) is indeed the eastern coastal side of Greenland between 75 and 79 ° N, making B17 an ideal site to capture the regional variability of local sea ice extent.

Similarly, the Baffin bay show high fluxes of Br release from its northern area (Toyota et al., 2011), probably due to its pronounced variability in winter/summer sea ice extent, resulting therefore in a significant formation of first year sea ice, which promotes *bromine explosions*.

As first approximation, we therefore believe that the B17 and B26 Bromine enrichment fluxes are modulated by the variability of the regional spring/summer sea ice extent of the proximal Greenland Sea and the Baffin Bay, respectively.

The ternary diagrams (fig. 7) seem to indicate that most of the sea-ice related signal can be accounted for considering only blowing snow as the main aerosol vector, starting to decline over time in good agreement with Br fluxes and Br_enr. A reduction of the sea ice cover would also reduce the $SO_4^{2-}/(Mg^++Ca^{2+}+K^+)$ depletion and bring the $Cl^-/Na^{2+}$ closer to 1.164 as a resulting of the increasing open water surface. Despite the slightly positive trend, it is difficult to characterize frost flowers as an important contributor to the sea-ice related chemical signal, as high and low values are more heterogeneously distributed over the years and the excess of both bromine and chlorine in recent years increases the values of the frost flower index F.F. A change in the ionic budget related to sea ice, however, might still be influenced on short time scale by changes in the transport mechanisms as well as the originating areas of air masses. For instance, a shift in sources from an area of strong seasonal sea ice variability to an area dominated by multi-year sea ice would produce a decrease in Bromine and Bromine enrichment as well as a relative increase in the $SO_4^{2-}/(Mg^++Ca^{2+}+K^+)$ ratio.

Nevertheless, Bromine records from ice cores clearly show the potential to represent the variability of regional sea ice extent from yearly to centennial scale. In conclusion, more ice cores sea-ice records are needed, not only in order to better represent the variability of sea ice extent in the Arctic region but to overcome regional effect by merging several profiles and to include chemical data deriving from different type of sea ice (first year vs multi-year sea ice).



## Acknowledgements

The Authors would like to thank the Helmholtz-group REKLIM, for providing financial support to Postdoc position of Damiano Della Lunga and for helpful comments and discussion, especially Ralph Tiedemann, Klaus Grosfeld, Martin Werner and Thomas Laepple, from Alfred Wegener Institute. The Authors would also like to thank Francois Burgay from
University of Venice, Italy, Sophie Darefuil from University of Grenoble, France, and Hubertus Fischer from University of Bern for providing help and assistance during the B17 CFA-melting campaign at University of Bern.

Tobias Erhardt and Camilla Jensen acknowledge the long-term support of ice-core research at the University of Bern by the Swiss National Science Foundation (SNSF) and the Oeschger Center for Climate Change Research.

The authors gratefully acknowledge the NOAA Air Resources Laboratory (ARL) for the provision of the HYSPLIT
transport and dispersion model and/or READY website (http://www.ready.noaa.gov) used in this publication.

## Authors contribution

All authors participated to the B17 measuring campaign by Continuous Flow Analysis at the University of Bern in 2017. BT performed all the ion Chromatography measurements on the B17 core at the Alfred Wegener Institute for Polar and Marine research. DDL performed all data analysis and wrote the manuscript, with help from MH, MB, TE and CMJ.
All authors commented and contributed to the final version of the text.

## Data availability

All data will be promptly uploaded on the data repository Pangaea.

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

**Tables and Figures**

**Table 1: Overview of the different details from the NGT ice cores utilized for this study.**

| | Core | | Reference |
|---|---|---|---|
| | B17 | B26 | |
| Location (°N; °W) | 75.25, 37.63 | 77.25, 49.22 | Weißbach et al., 2016 |
| Core length (m) | 100.8 | 88.4 [119.7 originally] | |
| Age range (Years AD) | 1363 – 1993 | 1502-1993 | |
| Acc. Rate (kg m⁻²a⁻¹) | 113-119 | 172-190 | |
| Elevation (m a.s.l.) | 2820 | 2598 | |
| Arrival Heights (m a.s.l.) | 2900[*] | 2650[*] | Cohen et al., 2007 |
| Year of Drilling (AD) | 1993 | 1995 | Schwager 2000 |
| Year of Analysis | 2017-2018 | 1996 | This study; Fischer 1997. |
| Time resolution (years) | 0.16 - 1 year | 1 | This Study |
| Depth resolution (cm) | 2 – 11 cm | ~ 15 cm | |
| Method of Analysis | C.F.A[**]. & I.C[***]. | I.C[***]. | Burgay et al., 2019; Fischer 1997; Schwager 2000 |

[*] Calculated as: elevation +0.5pblh, where pblh is height of average planetary boundary layer over central Greenalnd (see Cohen et al., 2007)

[**]Continuous Flow Analysis

[***]Ion Chromatography




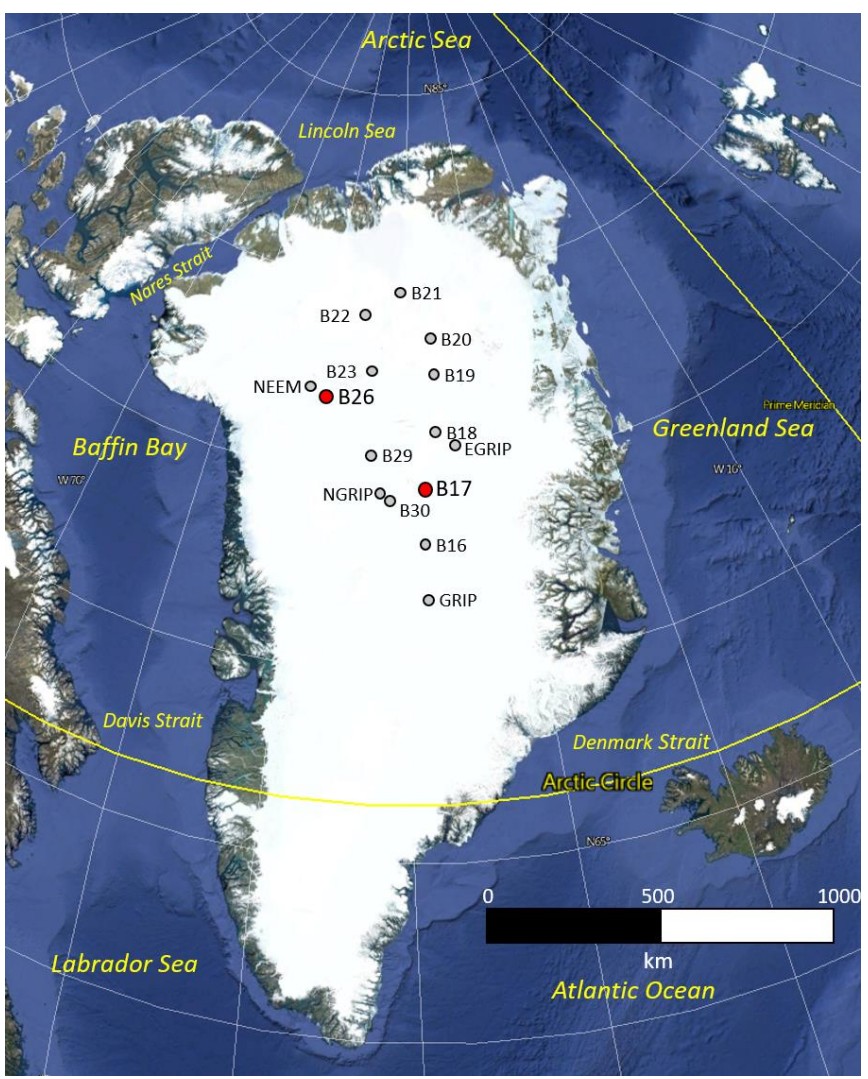

**Figure 1: Map showing the location of the NGT ice cores as well as other main Greenland drilling sites. B17 and B26 are marked in red. Image background: © Google Earth – Digital Globe 2019.**



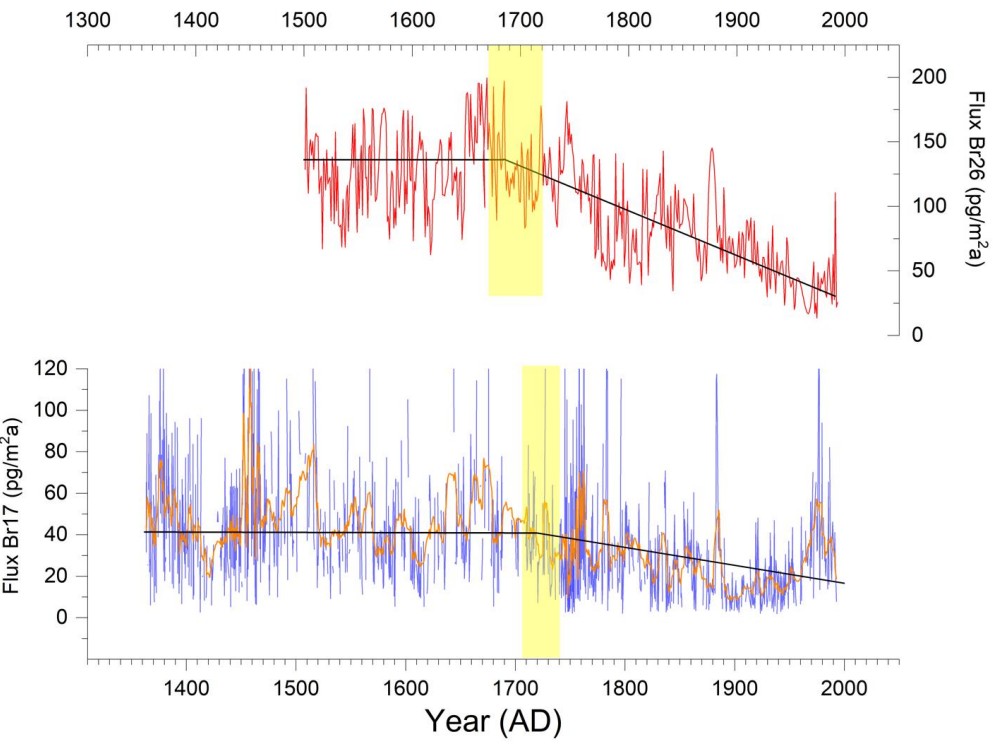


**Figure 2:** Bromine Fluxes (in picograms over square meters per year) for B17 and B26 ice cores. Black thick lines represent a RAMPFIT model fitted to data based on a initial guess (± 30 years) with 2000 simulations and boothstrap Confidence Intervals (yellow boxes), while orange line represent a 10-point adjacent averaging running mean of B17 original record.





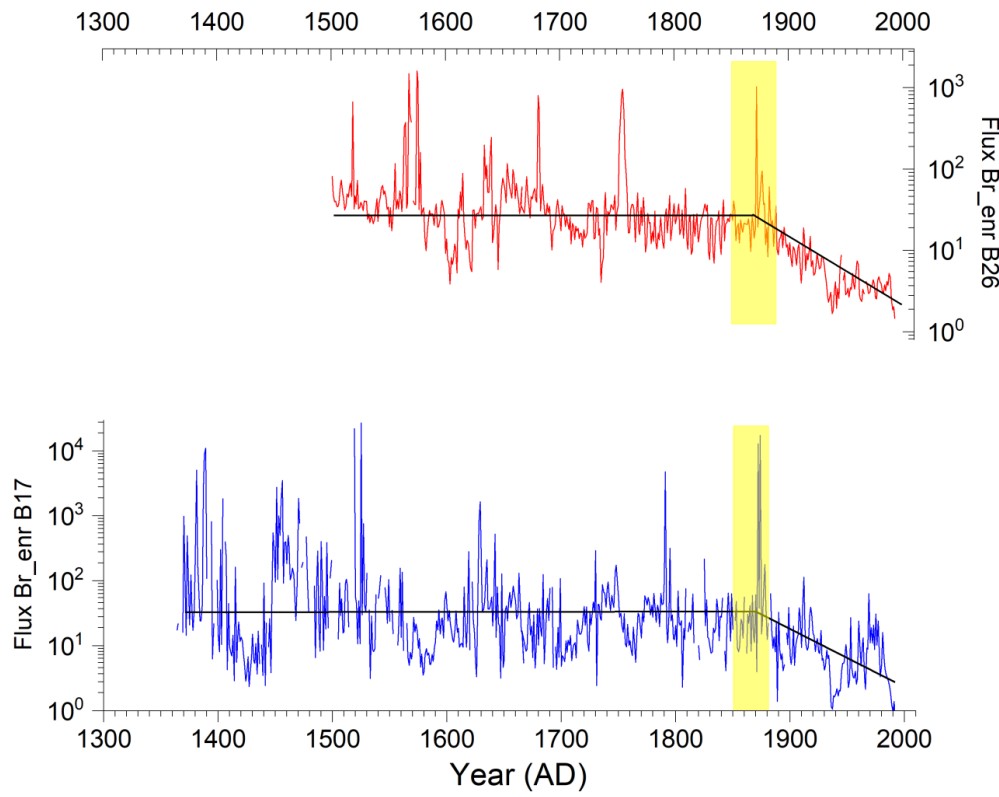

**Figure 3: Bromine_enrichment (adimensional) for B17 and B26 ice cores. Black thick lines represent a RAMPFIT model fitted to**
**data based on an initial guess (± 30 years) with 2000 simulations and boothstrap Confidence .Intervals (yellow boxes).**






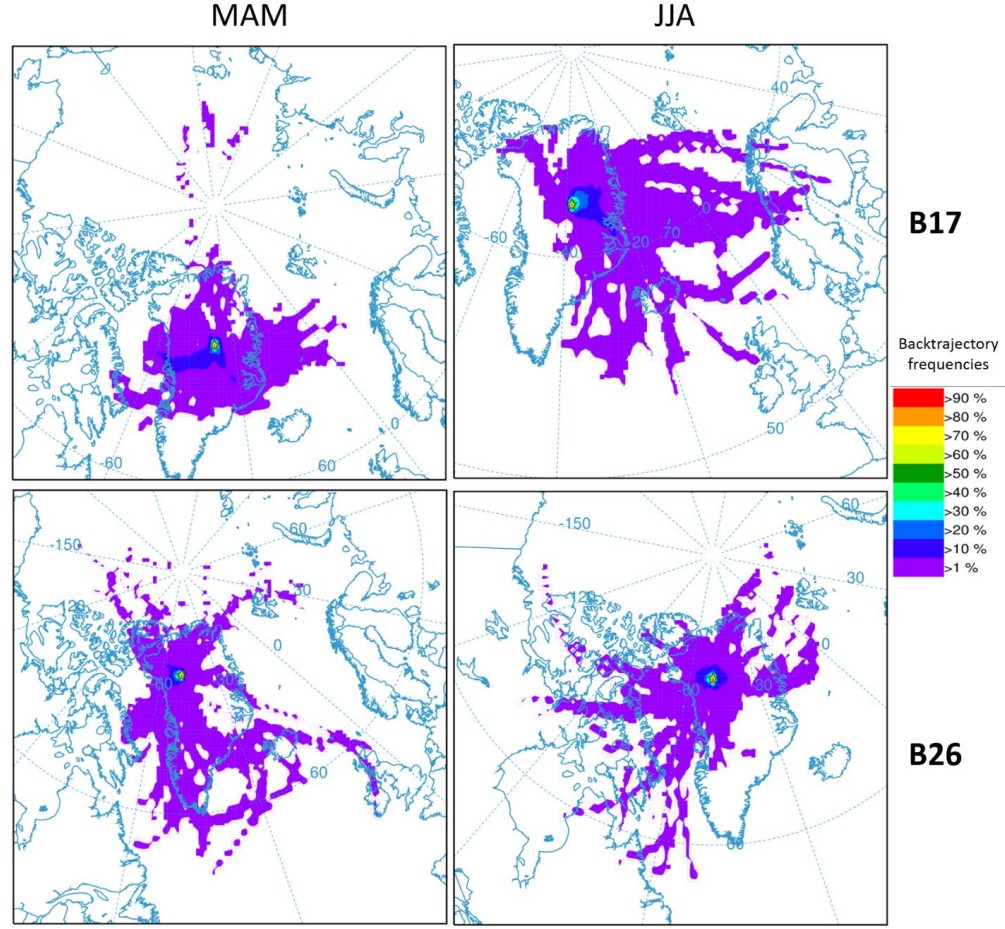


**Figure 4: Back-trajectories frequencies starting from B17 (top) and B26 (below) locations over the months of March-April-May (left) and June-July August (JJA) merged over the period 1950-1990 AD. Frequencies are expressed here as the number of trajectories passing through each grid square / number of total trajectories, where each trajectory is counted once per grid cell. Map generated with NOAA Hysplit 4 (April 2018 SVN 951).**




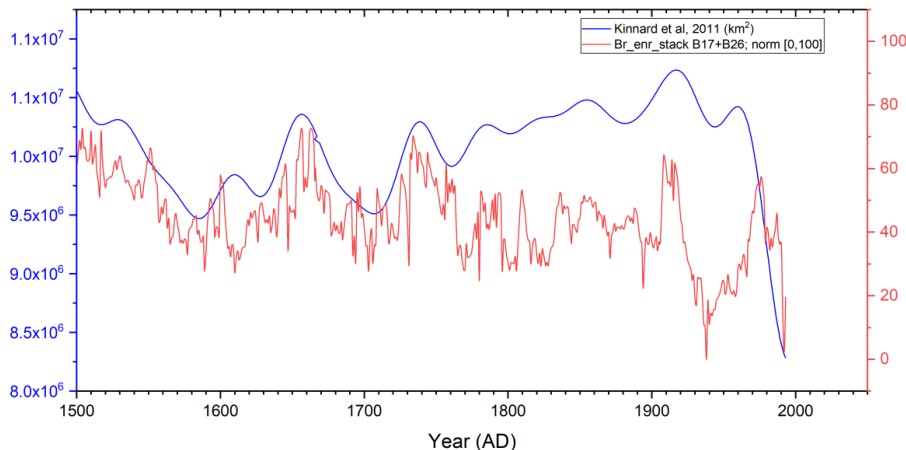

**Figure 5: Comparison of Normalized [0,100] stack of B17 and B26 Bromine enrichment (red) with the 40-year smoothed reconstructed late summer Arctic sea ice extent from Kinnard et al. (2011) in blue.**



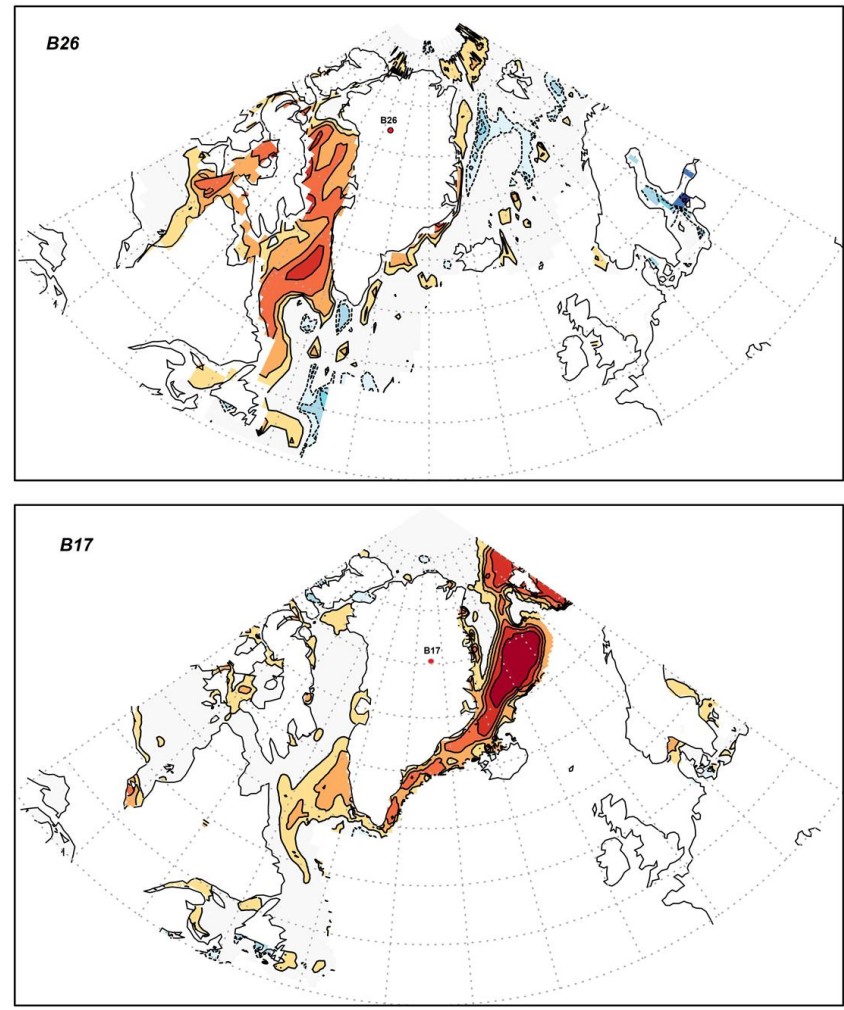

**Figure 6:** Correlation map of the times series of Bromine enrichment fluxes for B17 (top) and B26 (bottom) with HadISST sea ice
sea ice concentration on a 1-degree latitude-longitude grid between 60°N and 90°N between 1800 and 1993 AD. Correlation is
expressed as point-wise Pearson correlation [-1,1] coefficient. Maps generated utilizing ©KNMI Climate explorer (Trouet & Van
Oldenborgh, 2013)





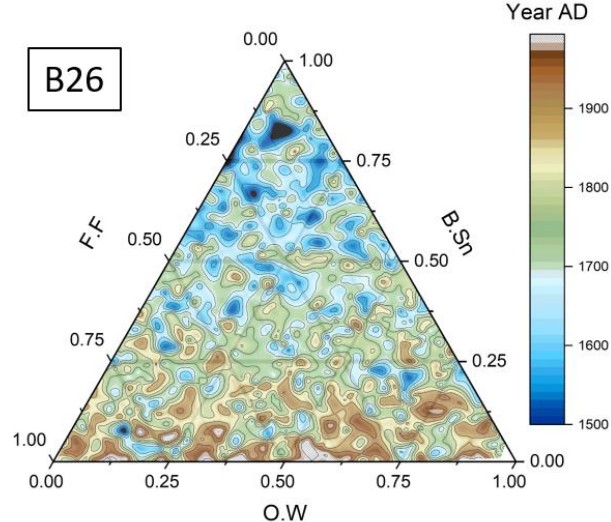

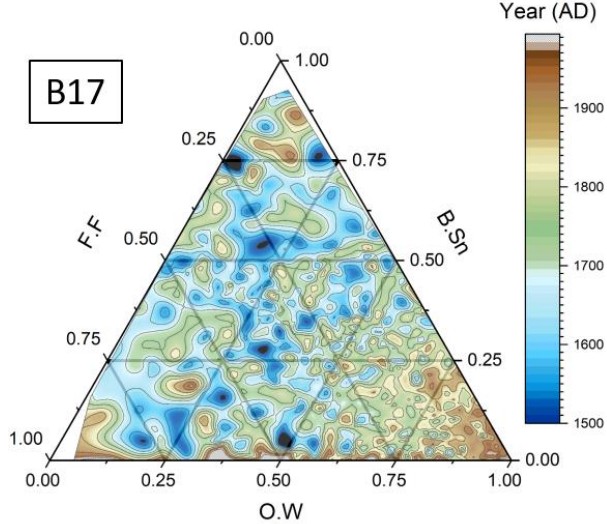

**Figure 7: Ternary diagrams relating Blowing Snow (B.Sn), Frost Flowers (F.F) and Open Water (O.W) chemical signatures in B26**
**(below) and B17 (up) ice core, expressed as contour maps, color-coded with time (Year AD).**



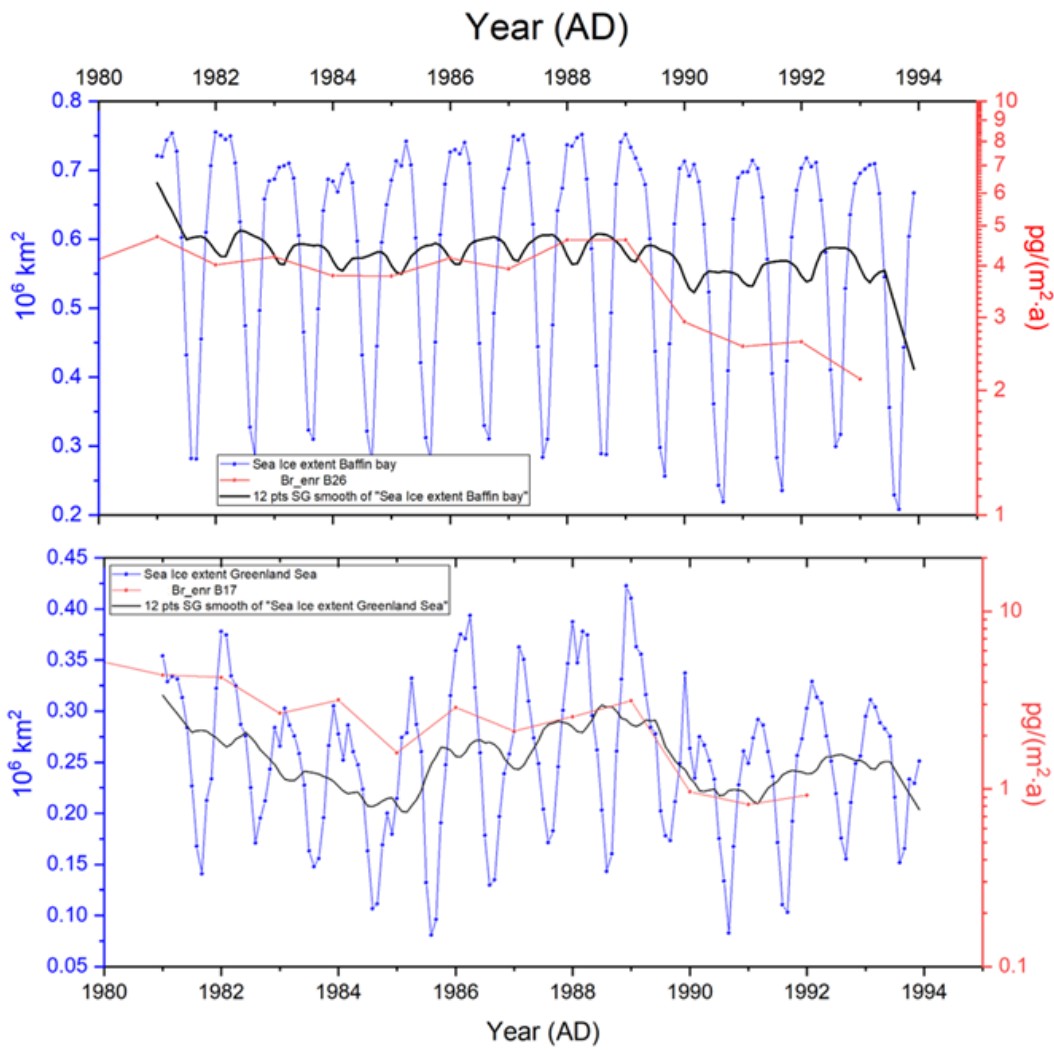

**Figure 8: Sea Ice extent (in millions of km²) for the Baffin Bay area (top) and Greenland Sea (below). Coloured lines are original data from NSIDC monthly sea ice concentration (Cavalieri et al., 1996), while solid black line represent a 20-point Savitzky-Golay moving average.**
