# Peer review of "Imprint of Arctic sea ice cover in North-Greenland ice cores"

_The Cryosphere, 2019_

## Referee Comment (RC1) · Anonymous Referee #1 · 4 Dec 2019

This paper correctly identifies the search for a new sea ice proxy as important. And it also identifies that ice cores, and ice core bromine in particular, have the potential to provide valuable information. However, Br in ice cores, like other potential sea ice proxies (Na, MSA) is difficult to interpret and requires a good understanding of atmospheric chemistry and deposition processes. This paper presents new datasets from two Greenland cores that include bromide data, and attempts to interpret the data in terms of sea ice extent and how it has changed over time. This is a promising undertaking but unfortunately in this case it has been ruined by a host of mistakes that should not have survived into the manuscript. It is impossible, given the number of erroneous calculations and assumptions, to assess whether the manuscript can be rescued. For that reason, the authors need to take it away and prepare a completely new and properly checked manuscript: this is not just a revision.

Here I present just the major issues, followed by a few of the more minor ones. By no means have I identified the numerous typographical and grammatical issues that also exist in the paper.

1. Section 2.2 gives a completely incorrect description of the Br explosion. The reactions shown here are those that cause tropospheric ozone depletion once Br has been activated. But Br in sea ice brines and related materials is present as Br- not as Br2, so the issue is how the bromide ion gets activated into gas phase species such as Br2 or BrO. The authors should refer for example to the review paper Simpson et al 2007 (Simpson, W. R. et al. Halogens and their role in polar boundary-layer ozone depletion. Atmos. Chem. Phys. 7, 4375-4418 (2007)), figure 4 and nearby reaction schemes (note this is a different paper to the one they already cite by Simpson).

2. A second issue with their interpretation is that on line 152 it is stated that Br and Br_enr are directly proportional to the extent of the surface available for the bromine explosion. However things are nowhere near as simple as this. Br can be transported inland both as bromide in sea salt aerosol and as HBr produced ultimately by the bromine explosion. As these will have different but unknown lifetimes, Br_enr will depend both on how much sea salt is produced and on the transport distance. This doesn't undermine the possible use of Br as a sea ice indicator, but does emphasise that it should not be presented in such a simplistic way.

3. On page 5, it is not clear to me why the authors have chosen to calculate the flux. Normally we would say that Greenland has a relatively high snow accumulation rate (compared to East Antarctica for example) where wet deposition will dominate. In such circumstances it is the concentration, not the flux, that more closely resembles the atmospheric concentration (although it must be said as a further complication that for a gas phase species such as HBr this might also not be so relevant). But in any case, the underlying logic of multiplying by the snow acc rate to calculate flux is the assumption that the same atmospheric concentration is diluted by higher snowfall rates. However, here the authors show that (weakly) the concentration in snow increases with snowfall rate, exactly the opposite of what we expect from this assumption. Given this, there is no reason to use fluxes, and what they are doing is inducing even higher values when snow accumulation rate is high.

4. In any case, it would definitely be important to show, which the authors don't, the raw concentration data, even if they choose eventually to also show fluxes. That allows the reader to decide. In fact they must have either a huge problem with the analysis or else they have calculated the fluxes incorrectly. They show (Figs 2 and 8) Br fluxes of 1-100 pg m^-2 a^-1, and snow acc rates (Fig S3) of 100-200 kg m^-2 a^-1. If I then back calculate the concentration as C=F/A, I get concentrations of less than 1 pg/kg, ie less than 10^-6 ng/g. And yet the detection limit is said to be 0.1 ng/g and previous authors have found

concentrations in Greenland around 1 ng/g. There seems to be an error somewhere of around 6 orders of magnitude.

5. I am also mystified that in Figure 2 we see values for site 26 for the recent past around 20-50 pg m^-2 a^-1, but in Fig 8 values for the same site are less than 5. There is a similar issue for site 17 though it is harder to see what the recent value in Fig 2 was. The authors should consider whether this is correct, and if so it requires some comment.

6. A consequence of using Br_enr is that if the concentration of Na increases but the amount of young sea ice (and hence Br explosion) stays the same, Br_enr will reduce (this statement depends on assumptions about the source of Na, but that is also part of the interpretation issue). The authors even mention this in line 246, but they don't then let us see what we need to see to judge it, which is the Na, Br, Br_enr and Br_exc for both cores. And they don't consider this in their interpretation. The strong (factor 2) increase of ssNa in FigS5 over the last century most likely dominates other causes of change in Br_enr at site B17, and certainly needs better discussion.

7. In section 2.5, the source partitioning is wrong and unjustified making everything in Figure 7 incorrect.
    a. Firstly there is no way to use sulfate depletion to diagnose anything once the anthropogenic era starts, because the introduction of anthropogenic sulfate in the last 200 years has completely overridden the sea ice signal.
    b. Secondly the authors introduce without explanation a distinction between frost flower chemistry and blowing snow (or snow on sea ice) chemistry. And yet everything that has been written would lead us to suggest that exactly the same processes of wicking and fractionation will occur in both cases. Mirabilite precipitation induces a large depletion of sulfate relative to Na and Cl, and a small depletion of Na relative to Cl; this is expected in both frost flowers and blowing snow (see eg Frey, M. et al (2019), First direct observation of sea salt aerosol production from blowing snow above sea ice, Atmos. Chem. Phys. Discuss., 2019, 1-53, doi:10.5194/acp-2019-259). Maybe the authors are suggesting that halite will also have been precipitated from frost flowers and not from blowing snow but I can't think of any reason why frost flowers a few cm above the (warm) water surface should have experienced colder temperatures than snow on sea ice that might be ten cm or more above the ocean surface; nor would this induce a fractionation of Na relative to Cl.

8. The paper has been carelessly assembled with numerous typographic errors. Even in the abstract there are several errors: bromine should not be capital-b throughout the paper; line 17 from not form; line 19 Arctic mis-spelt; line 28 Sea ice not Sea Ice. I won't go further but the paper needs really careful proofreading by more than one author.

9. A final major issue relates to the analytical description in the methodology in the supplement. The descriptions of the accuracy and precision are bizarre. Accuracy in analysis relates to the deviation of values from a traceable standard, but standards are not even mentioned in the definition given here. Precision is normally defined as the variation between successive determinations of the same sample, but the definition of precision given here seems to be somehow related to the Milli-Q blank (which one would hope would be close to zero concentration for most elements). Variation of the blank can be used to define a limit of detection but this has not been defined here. Finally typical precisions for ion chromatography are 5%, certainly not 0.05%.

It is a shame that so many problems are found as the correlation plots in Fig 6 look promising, but until we understand what the actual, correctly calculated concentration data show, it is hard to know whether the paper will be useful or not.

---

## Referee Comment (RC2) · Anonymous Referee #2 · 20 Jan 2020

The authors present new records of bromine and other trace ions from two Greenland ice cores, interpreting these records with respect to sea salt inputs and potential for reconstructing Arctic sea ice cover for the past 7 centuries. The primary metric for sea-ice reconstruction is bromine enrichment (Br_enr), a relative measure of bromine excess with respect to sea-salt bromine. The Br_enr records are compared to satellite observations for the last few decades, and to a pan-Arctic sea-ice reconstruction by Kinnard et al., for the last centuries. The work is a novel and helpful contribution to the understanding of bromine variability across Greenland and its potential utilisation as a sea-ice proxy. It is important to recognise that the authors present, and interpret, bromide (ion) and not (total) bromine concentrations, as the data were determined by Ion Chromatography. Although there is no specific reason to doubt the accuracy of the

measurements, very little analytical detail is provided and I will comment on this more below. Despite these shortcomings, the authors are to be commended for their novel and elegant approach to discrimination between open ocean, blowing snow and frost flower sea-salt sources, demonstrated in figure 7. The language is well-structured and understandable but with many minor grammatical errors.

Specific comments:

- It would have been helpful to have MSA included in this work, to complement previous comparisons between sea-ice proxies. If suitable MSA records are available they should be added to the interpretation.

- There are many small spelling and grammatical errors throughout the text. I provide some examples here: p.1 (Artic, trough/through), p.2 (pythoplankton), p.3 (residuals, twice than the one), p.5 (custumized), p.6 (Bromin), etc.

- Line 42, A centuries-long record of direct sea-ice observations is available for the north coast of Iceland.

- Line 60, add year to Kinnard citation

- Line 154-6. The authors should provide a reference or observational evidence to support the assertion that bromine explosions occur over multi-year sea ice. The study by Cox & Weeks instead indicates that surface salinity in multiyear sea ice is highly variable on metres-long length scales.

- Section 2.3. The "constrained" distribution of the back-trajectories in Figure 4 does not seem consistent with a 12-day run-time. Were the trajectories filtered for altitude? That is, were they constrained to pass through the Marine Boundary Layer at some point in time, to ensure they are more likely to sample the marine atmosphere? Please check that the run-time for each trajectory was not a shorter period (i.e., 2 or 3 days?) or some other form of data filter was not applied.

- Section 2.5. A typical test for the influence of mirabilite (representing frost flowers) is

the Na/SO4 ratio. Any reason why this approach was not adopted for this study?

- Line 318. Generic descriptions such as "a general increase" should be avoided. Please quantify how much and how significant this increase is.

- Table 1. Regarding the methods of analysis, please specify which parameters presented in the manuscript were produced by CFA (i.e., continuous flow analysis techniques such as Ca2+, Na+, Cl-, etc) and/or IC.

- Figures 2 and 3. It is very confusing and incorrect to write Br26 and Br17. Amend the text to something like 'B17 Brˆ- flux', etc. Furthermore the rampfit function also allows for error bounds to be applied to any change point. These should be added to the figure and quantified in text. Please consider combining Figures 2 and 3. Also consider including the sodium profiles for each core so the significance of any Sodium flux changes can be evaluated.

- Figure 4. Please ensure that the same geographical field is used in each panel, to make it easier for the reader to evaluate differences.

- Supplementary Methodology. Please quantify the LODs, sample concentration ranges and representative uncertainties for each of the ions presented in the text. From comparing this text and figure S1, it appears that the LoD for Br is 10 to 100% of the sample concentration?

- Figure S5. It is more helpful to plot something like decadal (or multi-decadal) averages, as it is unclear what time period a 20 point smoothing average represents.

- Figure S6. It would be interesting to plot this data together with Figure 5, to demonstrate larger-scale variability and consider other drivers of change in the polar atmosphere (e.g., background acidity, dust levels, etc).

---

## Editor Comment (EC1) · Jean-Louis Tison (Editor) · 27 Jan 2020

Dear Damiano and Co-authors,

First of all, I would like to apologize for the delay in handling your manuscript. It has been a great challenge to find appropriate reviewers that would accept to go through the manuscript and would not be involved too closely with the authors.

As you might have noticed from the reports, both reviewers were not very happy with the manuscript. One of them specifically recommended that the paper be rejected and totally rewritten. The other reviewer saluted some novel aspects but underlined the weaknesses of the methodology section (analytical techniques, back trajectories..) and the form, and therefore asked for major revisions.

I have been thinking a lot about my final decision , but I finally come to the conclusion that the best option would be to reject the paper at this stage and recommend a full revision along the lines of comments of both referees.

Good luck with it,

Jean-Louis Tison